# New methods for quantifying rapidity of action potential onset differentiate neuron types

**Ahmed A. Aldohbeyb**[1,2], **Jozsef Vigh**[1,3], **Kevin L. Lear**[1,4]*

**1** School of Biomedical Engineering, Colorado State University, Fort Collins, CO, United States of America,
**2** Department of Biomedical Technology, King Saud University, Riyadh, Kingdom of Saudi Arabia,
**3** Department of Biomedical Sciences, Colorado State University, Fort Collins, CO, United States of America,
**4** Department of Electrical and Computer Engineering, Colorado State University, Fort Collins, CO, United States of America

* kllear@engr.colostate.edu

**Data Availability Statement:** All relevant data are within the manuscript and its Supporting Information files.

**Funding:** The authors acknowledge support from the Colorado State University Libraries Open

## Abstract

Two new methods for quantifying the rapidity of action potential onset have lower relative standard deviations and better distinguish neuron cell types than current methods. Action potentials (APs) in most central mammalian neurons exhibit sharp onset dynamics. The main views explaining such an abrupt onset differ. Some studies suggest sharp onsets reflect cooperative sodium channels activation, while others suggest they reflect AP back-propagation from the axon initial segment. However, AP onset rapidity is defined subjectively in these studies, often using the slope at an arbitrary value on the phase plot. Thus, we proposed more systematic methods using the membrane potential's second-time derivative ($\ddot{V}_m$) peak width. Here, the AP rapidity was measured for four different cortical and hippocampal neuron types using four quantification methods: the inverse of full-width at the half maximum of the $\ddot{V}_m$ peak (IFWd$^2$), the inverse of half-width at the half maximum of the $\ddot{V}_m$ peak (IHWd$^2$), the phase plot slope, and the error ratio method. The IFWd$^2$ and IHWd$^2$ methods show the smallest variation among neurons of the same type. Furthermore, the AP rapidity, using the $\ddot{V}_m$ peak width methods, significantly differentiates between different types of neurons, indicating that AP rapidity can be used to classify neuron types. The AP rapidity measured using the IFWd$^2$ method was able to differentiate between all four neuron types analyzed. Therefore, the $\ddot{V}_m$ peak width methods provide another sensitive tool to investigate the mechanisms impacting the AP onset dynamics.

## Introduction

The initiation and propagation of action potentials (APs) are key processes of neural communication. Our understanding of the generation of APs advanced greatly by using the Hodgkin and Huxley (HH) model of AP generation. It states that an AP is generated in the giant squid axon due to rapid discharging and recharging of the axon membrane by ionic sodium and potassium currents through a single type of membrane channel for each ion [1]. However,

Access Research and Scholarship Fund for partial coverage of publication charges. Ahmed A. Aldohbeyb acknowledges support from King Saud University for his PhD studies. Jozsef Vigh acknowledges support from Colorado State University, College of Veterinary Medicine and Biomedical Sciences Research Council Award and National Eye Institute.

**Competing interests:** The authors have declared that no competing interests exist.

**Abbreviations:** AIS, axon initial segment; AP, action potential; PC, pyramidal cell, PVBC, parvalbumin-positive basket cell; RS, regular-spiking cell; FS, fast-spiking; VGSC, Voltage-gated sodium channel; VGKC, Voltage-gated potassium channel; $V_m$, membrane potential; $\ddot{V}_m$, the second-time derivative of $V_m$; IFWd$^2$, the inverse of full-width at the half maximum of the $\ddot{V}_m$ peak; IHWd$^2$, the inverse of half-width at the half maximum of the $\ddot{V}_m$ peak; HH model, Hodgkin–Huxley model; CLES, common language effect size; RSD, relative standard deviation.

subsequent investigations, aided by continuous improvement of imaging and measurement techniques, clarified that APs recorded from neuronal somas are more complicated than those in the giant squid axon [2]. For example, typical neurons in mammals express voltage-gated ion channels (VGICs) that are permeable to calcium ions in addition to the sodium and potassium channels. Furthermore, the same neuron might express more than a single type of membrane channel for each ion; channels containing various subunits result in differences in biophysical and pharmacological properties [2–4]. The presence of various VGIC sets across different neurons causes the shape of the AP to vary significantly in the same animal [5]. This variability in ion channels and the resultant APs adds to the complexity of understanding the role of each type of VGIC in neuronal firing behavior.

Nevertheless, the role of voltage-gated sodium channels (VGSCs) in AP generation is thought to be well defined: the depolarization-induced opening of VGSCs is a critical step in AP initiation. The gating properties of VGSCs of mammalian central neurons are considered to be similar to those in the giant squid axon, including the notion that individual VGSCs open independently upon membrane depolarization [2]. However, several studies in cortical neurons reveal that actual AP initiation appears faster than classical HH-type models predict [6–8]. Some studies suggested that a more complex gating property of VGSCs (i.e., "cooperative gating") could be responsible for the deviation from the classic HH-type models [6–8], whereas others suggested that the discrepancies could be explained by a multicompartmental HH model in which AP backpropagation from the axon can alter the AP onset rapidity in somatic recordings [9]. Several other studies suggested that the rapidity of AP onset is influenced by resistive coupling between the axon and soma [10, 11], and by the size of the dendritic tree [12]. Nonetheless, the ongoing debate on AP initiation mechanisms demonstrates the importance of the topic [6–15].

Notably, the method for quantifying AP onset rapidity differs from one study to another. One common approach, the phase-slope method, evaluates the slope of the phase plot, i.e., the first time derivative of the membrane potential ($\dot{V}_m$) as a function of the potential ($V_m$), at arbitrary values of $\dot{V}_m$ (ranging from 5 to 50 mV/ms) termed the "criterion level" [6, 10, 16]. Such arbitrary choices of criterion level can confuse and complicate the comparison between different models and experimental data across studies. Another way to analyze AP onset is the error ratio method, which has been shown to quantitively differentiate fast-onset APs in rats' cortical neurons from slow-onset APs in snail neurons [17]. Recently, we have shown that the full width at half maximum (FWHM) and the half-width at half maximum (HWHM) of the rising phase of the membrane potential's second-time derivative ($\ddot{V}_m$) provides systematic and consistent methods to quantify the rapidity of AP onset [18]. Here, we compare these $\ddot{V}_m$ peak width methods [18] to the phase slope [6, 9, 10] and the error ratio method [17] via analysis of onsets of APs recorded from cortical and hippocampal neurons. The results suggest that the two $\ddot{V}_m$ peak width methods of quantifying AP onset rapidity more robustly distinguish between somatosensory cortical neurons and hippocampal neurons in mice than the phase plot slope and error-ratio methods. Overall, we propose that the two $\ddot{V}_m$ peak width methods are sensitive and robust methods to differentiate neuron types based on AP rapidity, and hence might be used as a classification parameter across APs.

## Methods

### Data source for AP recordings

Electrophysiological recordings were obtained from two databases. Recordings from the somatosensory cortex were obtained from the GigaScience database [19], whereas recordings

from hippocampal neurons were obtained from the CRCNS database [20]. For the somatosensory cortical recordings, the experimental procedures and data are found in da Silva Lantyer *et al.* [21]. The analyzed cortical data were from current-clamp recordings of pyramidal regular-spiking (RS) neurons (n = 27) and fast-spiking (FS) neurons (n = 7) in layers (L2/3) of the primary somatosensory cortex in adult mice. These recordings were obtained and uploaded to the database by Angelica da Silva Lantyer (AL) and were found to be the lowest noise recordings in that database. The recording labels in the database are given in da Silva Lantyer et al. Supporting Information [19]. For the hippocampal neurons, the experimental procedures and recordings are found in Lee *et al.*, [20, 22]. These current-clamp recordings were made from adult mice hippocampal CA1 neurons. The recordings analyzed here are from 17 RS pyramidal neurons and 6 FS interneurons. The RS pyramidal neurons were further divided into two groups: neurons located in the CA1 superficial sublayer are labeled as superficial pyramidal cells (sPCs) (n = 8), and neurons located in the CA1 deep sublayer are labeled as deep pyramidal cells (dPCs) (n = 9). Finally, the 6 FS hippocampal interneurons were identified as 6 parvalbumin-positive basket cells (PVBCs).

Recordings included in our analysis had to satisfy criteria regarding numbers and spacing of APs: each current step must have contained at least 2 APs with an interspike interval that was at least 30 ms for RS pyramidal neurons and 12 ms for fast-spiking neurons. The 30 ms limit between RS neurons' APs was set to exclude the variability caused by incomplete deinactivation of the sodium channels [6]. This limit did not exclude many APs since RS neurons firing rate is 32 ± 7 Hz, whereas such a limit can exclude up to half the APs from FS neurons, which have a higher firing rate of 61 ± 9 Hz [23]. Thus, for FS neurons, the lower limit on the interspike interval between APs was set to be 12 ms. That value was chosen because, within the data analyzed here, it was the minimum interval between APs needed to calculate the error-ratio, which requires the AP trace to be fit starting 5 ms to 10 ms before AP onset [17]. The number of AP spikes that satisfied the interspike interval criteria ranged between 58 to 222 APs for each pyramidal cortical neuron recording, between 210 to 514 APs for each FS cortical neuron, between 103 to 182 for each pyramidal hippocampal neuron, and between 80 to 199 for each PVBC. Then, for each AP that fulfilled the above criteria $\dot{V}_m$ and $\ddot{V}_m$ were computed using MATLAB's *diff* function and then interpolated to a resolution of $\Delta t = 1$ μs using MATLAB's *spline* function unless stated otherwise in the Results section (MATLAB V9.5 (R2019b)).

## Quantification of rapidity of AP onset

The rapidity of AP onset was determined using four methods: the inverse FWHM of the $\ddot{V}_m$ peak (IFWd$^2$) [18], the inverse HWHM of the $\ddot{V}_m$ peak (IHWd$^2$) [18], the slope of the curve in the AP phase plot [6, 9, 10], and the error ratio method [17]. The first three methods measure onset rapidity in units of inverse time, while the last method yields a dimensionless value. The IFWd$^2$ and IHWd$^2$ methods were described previously in [18]. In short, the initial portion of the rising phase of $\ddot{V}_m$ versus time was selected from 3 ms before the AP peak, although this starting point is not critical for the IFWd$^2$ and IHWd$^2$ calculation, up to the time when $\ddot{V}_m$ drops to zero after peaking. The FWHM and HWHM quantified from the selected $\ddot{V}_m$ portion and the width values, with units of time, were inverted to obtain rapidity. The FWHM was determined to be the time difference between when $\ddot{V}_m$ rises past half the maximum value and when it decays below half the maximum value as shown in Fig 1C. The HWHM was similarly calculated, except the HWHM was defined as the time it takes $\ddot{V}_m$ to rise from half its maximum value to its maximum value (Fig 1C). For the phase slope method, the rapidity was

quantified exactly as described in previous studies [6, 9, 10]. The slope of the tangent line in the phase plot at the criterion level of $\dot{V}_m$ = 10 mV/ms defines the phase slope rapidity of AP onset for this study, unless a different criterion level is stated (Fig 1B).

The final method for quantifying AP onset rapidity was the error-ratio method, which was introduced by Volgushev *et al*. [17]. It was defined as the ratio of the errors for an exponential fit to a two-segment piecewise linear fit (Fig 1D). Volgushev *et al*. specified the fitted portion of the AP trace to be 5–10 ms before the AP onset to the point when either $\dot{V}_m$ reaches 20 to 30% of its maximum value or $V_m$ is 3 to 10 mV above the AP onset voltage. In this study, we choose the 30% $\dot{V}_m$ maximum value as the upper limit unless stated otherwise. The 30% $\dot{V}_m$ maximum value gives enough data points above the onset voltage to capture the upward increase of the AP trace in the phase plot before the rightwards curving of the AP trace toward the maximum value. The ratio of the exponential fit error to piecewise linear fit error can determine if the AP onset is fast or slow. An error-ratio higher than 3 indicates a sudden, fast onset best fit by two piecewise linear segments, while a ratio below 2 indicates a slow continuous onset best fit by a smoothly increasing exponential curve. Following Volgushev *et al.'s* paper, the data were interpolated using MATLAB *spline* interpolation function. The fitting functions were implemented as described in reference [17], except all three exponential parameters were obtained from a single fitting procedure using the MATLAB *fit* function with a starting estimate for the three parameters rather than the two-step fitting process described by Volgushev *et al*.

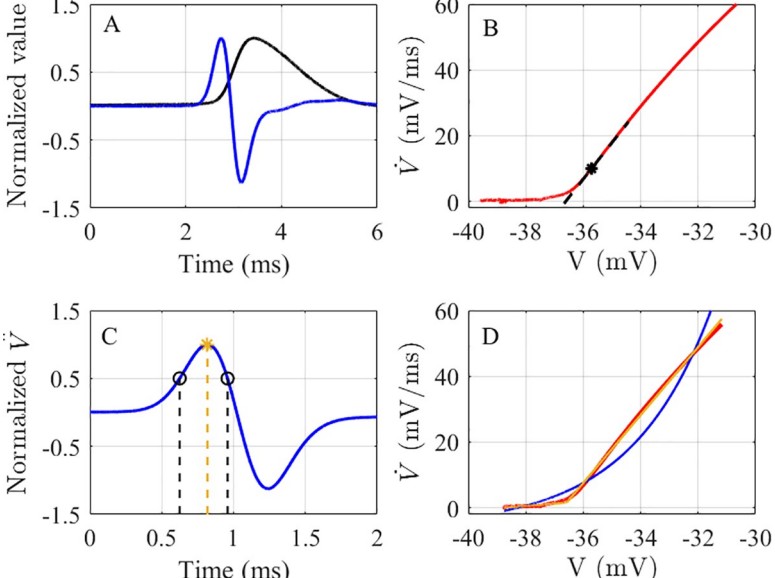

**Fig 1. The AP rapidity quantification methods. A)** A normalized AP from a mouse RS somatosensory cortical neuron (black), and its normalized $\dot{V}_m$. **B)** The initial portion of the phase plot of the same AP in A. The black asterisk indicates the criterion level (10 mV/ms), and the dashed black line represents the phase slope. **C)** The normalized $\ddot{V}_m$. of the same AP in A on a smaller time scale. The black circle indicates the points at which $\ddot{V}_m$. reaches half its maximum value before and after peaking, while the yellow asterisk represents the maximum normalized $\ddot{V}_m$. value. The time between the black dashed lines is the FWHM, and the time between the first black dashed line and the yellow dashed line is the HWHM. The IFWd² and IHWd² are defined as the reciprocal of the FWHM and HWHM, respectively. **D)** The red line shows the selected portion of the phase plot in A. The blue line shows the exponential fit, and the yellow line shows the piecewise linear fit of the selected portion of the phase plot. The error ratio is calculated as the ratio of errors of exponential fit to the piecewise-linear fit.

## Other AP parameters

In addition to analyzing the rapidity of each AP, the onset voltage, amplitude, and width of each AP were also analyzed. The AP onset voltage was taken to be the voltage at which $\dot{V}_m$ reaches 10 mV/ms as defined by Naundorf *et al.* [6]. The AP amplitude was measured as the difference between the membrane voltages at AP onset and the peak voltage, and the spike width was measured as the full width at half the AP amplitude.

## Statistical analysis

The mean and standard deviations of rapidity of multiple APs from multiple neurons of the same type were calculated using two statistical approaches: neuron-level pooled statistics using all APs (58–514 per neuron) meeting the selection criteria and conventional statistics on the combined first 50 APs from each neuron. Pooling combines the means or variances for each neuron of a particular type by weighting them by the number of selected APs of each neuron [24]. For conventional statistics, each neuron of a given type was effectively weighted equally in that each one contributed 50 APs to the combined sample and no mean or standard deviation was computed for individual neurons. After calculation of the pooled and conventional means and standard deviations, the relative standard deviation (RSD), which is defined as the ratio of the standard deviation to the mean for each neuron type was calculated using the values from the two statistical approaches. Pairs of neuron types were then compared using two-tailed Student's t-tests, the Mann-Whitney U test, and effect size using two test methods; Cohen's d and common language effect size (CLES) [25]. While both the t-test's t-score and the Mann-Whitney U-test z-score are enhanced by the large AP sample sizes (hundreds of spikes), the effect size is not. Thus, the first two reflect statistical differences in the mean rapidity of neuron types, and the latter one provides an indication of the ability to classify individual neurons.

## Results

### Comparison between the AP rapidity quantification methods

**Somatosensory cortical neurons.** The recordings from cortical neurons were analyzed to determine the mean and standard deviation of AP rapidity using the four quantification methods for each RS and FS neuron's spike train; values are plotted in Fig 2. When comparing the mean rapidity for each type of neuron of, the $IFWd^2$ and $IHWd^2$ methods show the lowest variation within each neuron type compared to the other methods. The cell-to-cell conventional RSD of the spike train rapidity measured using the $IFWd^2$ method was 22.2% for RS neurons and 14.7% for FS neurons. Corresponding RSDs using the $IHWd^2$ method were 18.2% and 15.7%, respectively. The phase slope method gave higher relative variation with RSDs of 33.4% and 16.7%, respectively. The error-ratio method produced yet higher variation across the same neuron type with 104% and 81.4% RSDs, respectively, for RS and FS neurons. Table 1 summarizes the conventional mean and standard deviation of each rapidity quantification method as well as other electrophysiological properties for RS and FS neurons. Furthermore, the choice of either conventional or pooled statistics did not alter the basic results. Using the pooled statistics, the $\ddot{V}_m$ peak width methods still show the smallest relative variation among the AP rapidity calculations methods (S1 Table).

**Hippocampal neurons.** The analysis of the hippocampal neuron recordings also showed that the $IFWd^2$ and $IHWd^2$ methods had much lower RSD than either the phase slope or error ratio methods, as seen in Fig 3. Also, as shown in Table 2, despite the slightly lower rapidity of superficial pyramidal neurons compared to the deep pyramidal neurons, none of the methods

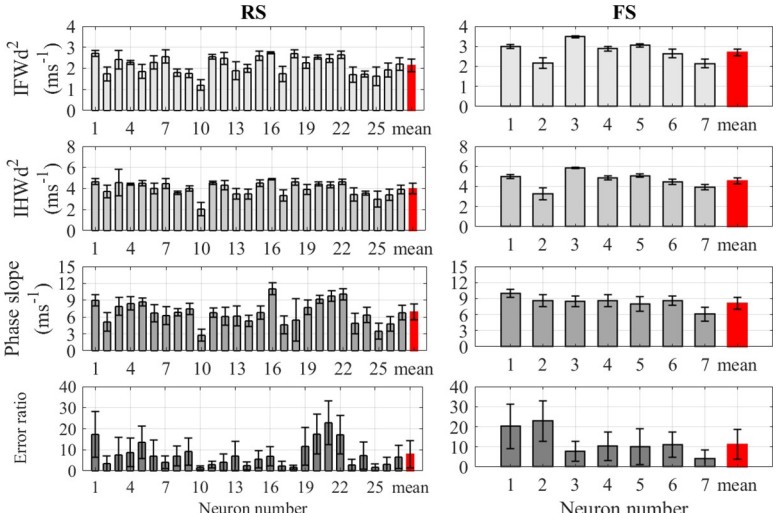

**Fig 2. The AP rapidity calculated using the four quantification methods for cortical neurons.** Comparison between the rapidity quantification methods for RS pyramidal cortical neurons (left) and FS cortical neurons (right). The units for IFWd², IHWd², and phase slope are in ms⁻¹ while the error-ratio is dimensionless. Error bars indicated standard deviations for each neuron. The red bar at the end of each figure indicates the pooled mean value for all neurons and its error bar indicates the pooled standard deviation.

shows a statistically significant difference between superficial and deep neurons using pooled statistics. Thus, all the hippocampal pyramidal neurons, despite their varying location, were treated as the same cell population (Table 1).

**Table 1. Electrophysiological properties using conventional mean and standard deviation.**

| | Cortex | | Hippocampus | |
|---|---|---|---|---|
| | **RS** | **FS** | **RS** | **FS** |
| | **(pyramidal)** | | **(pyramidal)** | **(PVBCs)** |
| **Number of neurons** | 27 | 7 | 17 | 6 |
| **IFWd²** | 2.2 ± 0.5 | 2.9 ± 0.4 | 4.7 ± 0.8 | 6.8 ± 0.5 |
| **(ms⁻¹)** | | | | |
| **IHWd²** | 4.1 ± 0.7 | 4.8 ± 0.8 | 7.9 ± 1.9 | 12.0 ± 1.1 |
| **(ms⁻¹)** | | | | |
| **Phase Slope** | 7.1 ± 2.4 | 8.6 ± 1.4 | 70.7 ± 103.5 | 12.6 ± 7.7 |
| **(ms⁻¹)** | | | 46 ± 19.4 [a] | 13.6 ± 7.5 [a] |
| **Error ratio** | 8.7 ± 9.0 | 13.2 ± 10.8 | 7.7 ± 3.2 | 0.8 ± 0.6 |
| **(dimensionless)** | | | 7.4 ± 3.4 [b] | 8.6 ± 3.2 [b] |
| **Amplitude** | 64.3 ± 12.3 | 61.1 ± 3.9 | 72.7 ± 7.1 | 48.3 ± 6.7 |
| **(mV)** | | | | |
| **Width** | 1.9± 0.6 | 0.7 ± 0.1 | 1.3 ± 0.2 | 0.3 ± 0.03 |
| **(ms)** | | | | |
| **Onset potential** | -28.3± 9.9 | -39.2± 7.1 | -33.4 ± 3.4 | -33.1 ± 2.4 |
| **(mV)** | | | | |

All data expressed as mean ± SD.

[a] using piecewise cubic interpolation.

[b] the upper limit was set to 3 mV above the AP onset. The number of APs used to obtain the mean and SD values are 1350 for RS cortical neurons, 350 for FS cortical neurons, 850 for RS hippocampal neurons, and 300 for FS hippocampal neurons.

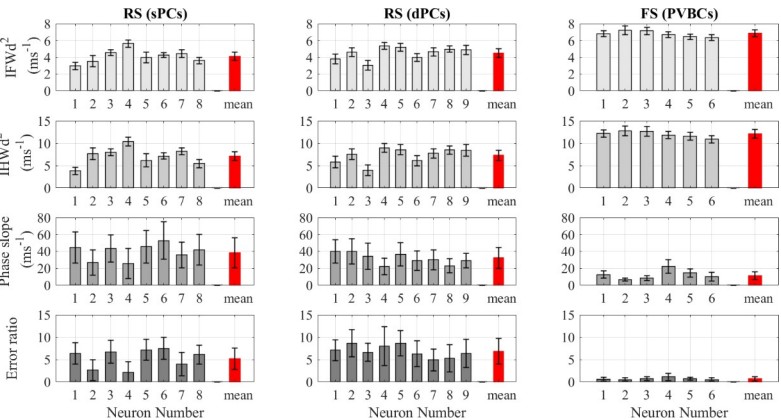

**Fig 3. The AP rapidity calculated using the four quantification methods for hippocampal neurons.** Comparison between the onset rapidity quantification methods applied to hippocampal neuron APs. All the values here are obtained using the spline interpolation function except for phase slope values, which were obtained using the pchip interpolation function.

The hippocampal neuron's conventional RSD of the spike train rapidity measured using the $IFWd^2$ method was 17.4% for RS pyramidal neurons and 8.0% for FS PVBCs. Corresponding RSDs using the $IHWd^2$ method were 24.4% and 9.6%. The error ratio method had a high relative variation with RSDs of 42.0% and 73.3%, respectively. For the hippocampal neurons, the phase slope method produced the highest variation with 147.9% and 61.2% RSDs, respectively, for RS pyramidal neurons and FS PVBCs. Strikingly, the $\ddot{V}_m$ peak width methods indicate that the FS PVBC have more rapid onset than the RS neurons, similar to the cortical neuron results, while the error ratio and the phase slope methods gave the opposite relation. The error ratio's much smaller mean and SD values for FS hippocampal neurons can be attributed to the data selection, which is discussed in a following section. Also, the high variation in the phase slope value was due to low sampling rates, motivating the adoption of a different interpolation function for further analysis using the phase slope method as discussed in the next section.

**Impact of sampling rate and interpolation method.** Sampling rates used during data acquisition and interpolation formulas impact AP onset rapidity, especially when quantified with the phase slope or error ratio methods. For hippocampal RS pyramidal neuron recordings, the phase slope method produced high standard deviations, which can be attributed to the sampling rate of the recordings and the interpolation function. The sampling rate for the hippocampal recordings was 10 kHz, giving one data point every 100 µs. In contrast, the sampling rate for the cortical recordings analyzed in this paper was 20 kHz. The lower sampling rate (i.e., the longer time between consecutive samples of $V_m$ in hippocampal recordings)

**Table 2. The mean and standard deviation of rapidity for RS hippocampal neurons for each quantification method using pooled statistics.**

| Interpolating function | Cell type | $IFWd^2$ | $IHWd^2$ | Phase slope | Error ratio |
|---|---|---|---|---|---|
| | | $(ms^{-1})$ | $(ms^{-1})$ | $(ms^{-1})$ | (dimensionless) |
| **Spline** | Pyramidal (n = 9) | 4.5±0.5 | 7.3±1.2 | 43.3±38.9 | 6.8±2.9 |
| | (deep) | | | | |
| | Pyramidal (n = 8) (Superficial) | 4.1±0.5 | 7.1±1.0 | 53.4±86.7 | 5.2±2.4 |
| **Piecewise cubic** | Pyramidal (n = 9) | 4.3±0.4 | 7.2±1.7 | 32.2±12.4 | 8.1±4.0 |
| | (deep) | | | | |
| | Pyramidal (n = 8) (Superficial) | 4.0±0.5 | 7.4±2.2 | 38.5±17.8 | 6.55±3.8 |

results in only a few data points during the rising phase of the AP. Although such a lower sampling rate is acceptable to reconstruct most AP details with high fidelity, high precision analysis of AP onset dynamics benefits from faster sampling rates and accurate interpolation. Consequently, the choice of interpolation function can impact the value of the AP rapidity, especially when evaluating the phase slope at a specific criterion level in the phase plot.

Spline interpolation can cause significant excursions, i.e., bowing between two measured data points resulting in a huge variation in the phase slope rapidity from one AP to another in the same spike train. Such excursions were apparent in 3 pyramidal neuron recordings and caused the high standard deviation value for the phase slope method given in Table 1. Thus, the piecewise cubic interpolation function (the function *pchip* in MATLAB R2019b) was used to interpolate the pyramidal hippocampal neuron recordings. Table 2 shows a comparison of the mean and standard deviations of AP onset rapidity for the two interpolation methods in combination with each rapidity quantification method. Using the cubic piecewise interpolation function substantially reduced both the mean and standard deviation values obtained using the phase slope method. Such a significant reduction in the phase slope values is expected since the slope is calculated at a specific criterion level (at 10 mV/ms) of the dependent (vertical axis) variable in phase space plots. As soon as an interpolated value of the dependent variable, $\dot{V}_m$, is at or above the chosen criterion level, the neighboring data points are used to find the tangent line (i.e. the slope) on the phase plot. This can occur prematurely if the spline interpolation reaches dependent values greater than either data point. Unlike the spline interpolation, the cubic piecewise (*pchip*) interpolation never produces an excursion beyond the data points between which it is interpolating, and hence the variation in the phase slope values decreased when *pchip* interpolation was employed. However, for the error-ratio method, *pchip* interpolation increased the mean and standard deviation of AP rapidity values by more than 40% compared to spline interpolation. This increase in the error ratio can be primarily attributed to the increase in the exponential fit error. Unlike the cubic piecewise interpolation function, the cubic spline interpolation function has a smooth transition between the points, which causes a better overlap between the interpolated AP portion and the exponential fit. Hence, the mean square error will be lower with spline interpolation than that with cubic piecewise interpolation.

In contrast, the IFWd$^2$ and IHWd$^2$ mean values are minimally affected by the choice of the interpolation function with less than a 5% difference. While changing the interpolation functions altered the standard deviation of rapidity for the IFWd$^2$ method by less than 1%, the IHWd$^2$ standard deviation roughly doubled. The higher variation in IHWd$^2$ values occurs because, in a few recordings, the peak of $\ddot{V}_m$ can be missed and partially truncated by the piecewise cubic function (S4 Fig). Regardless of the interpolation function, the IFWd$^2$ and IHWd$^2$ methods show consistent mean values and smaller variation across the same neuron type.

**Impact of phase slope criterion level.**    The rapidity determined by the phase slope method appears to depend on the criterion level at which it is calculated [10, 15]. Typically, the criterion level should be set to be in the linear region just above "the kink" and higher than the baseline noise [6, 9]. If onset is characterized by a phase plot that is linear over a wide range of $\dot{V}_m$ values, changing the criterion level will not significantly alter the phase slope and thus the rapidity. However, if the phase plot is non-linear or has multiple linear segments just above AP onset, or the criterion level was set very close to the onset kink, then the phase slope value will depend on the chosen criterion level. For example, varying the phase slope criterion level from 10 to 40 mV/ms resulted in less than a 20% change in mean rapidity for hippocampal RS pyramidal neurons. Such a small change is expected since the phase plot is nearly linear over the chosen range, as shown in Fig 4B for a typical case. However, for the hippocampal FS PVBCs,

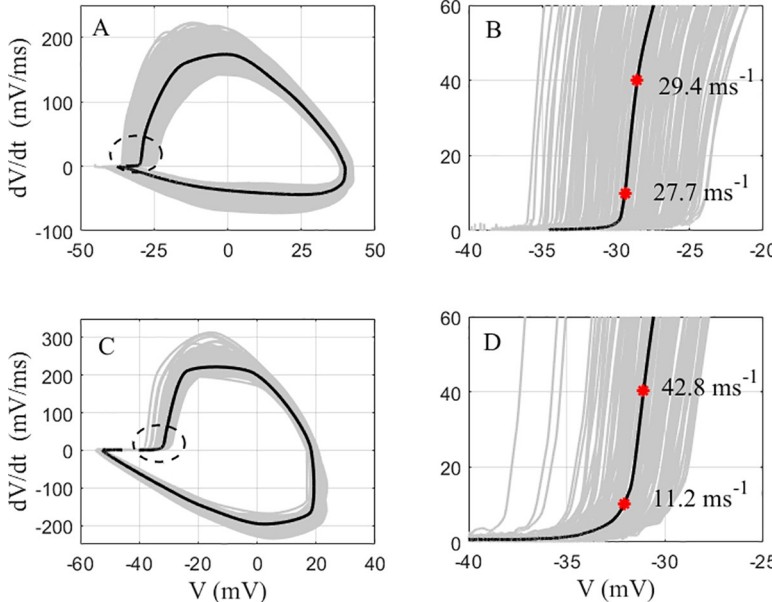

**Fig 4. Comparison of the impact of the phase slope criterion level between a hippocampal pyrmidal neuron and a hippocampal PVBC. A:** A cumulative phase plot of the APs from a hippocampal pyrmidal neuron (gray), and their average (black). **B:** The portion of the phase plot of A near onset. **C:** A cumulative phase plot of the APs from a hippocampal PVBC (gray), and their average (black). **D:** The portion of the phase plot of C near onset. The red astericks in C and D represent the points at which the average $\dot{V}_m$ reached 10 mV/ms and 40 mV/ms, and the values next to the astericks are the phase slopes at those points.

varying the $\dot{V}_m$ criterion level caused the phase slope to increase significantly, as shown in Fig 4D. The 10 mV/ms criterion level is near the transition between the baseline and the vertical rise of the phase plot. Thus, the PVBCs phase slope measured at 40 mV/ms is almost 4 times larger than the value measured at 10 mV/ms (S5 Fig). Nonetheless, when the phase slope is measured in a single linear range, for example above 25 mV/ms for the FS PVBCs, the phase slope values are more consistent. Notably, changing the criterion level reversed the relative rapidity between the hippocampal RS pyramidal neurons and the FS PVBCs (S5 Fig). In contrast, the mean rapidity of AP onset of cortical FS neurons was always greater than the mean rapidity of RS neurons as quantified by the phase slope method over the full 10 to 40 mV/ms criterion level range.

**Impact of data selection limits on error ratio method.** Volgushev *et al.* used the error ratio to distinguish between rapid rat cortical neurons (error ratio > 3) and slow onset snail neurons (error ratio < 2), but a few percent of the neurons in their analysis would be incorrectly categorized by the error-ratio method. They found 49 cortical neurons had an average error ratio of 8.46 ± 3.87 with all neurons except one having a value above 3 [17]. In contrast, 29 snail neurons had an average error ratio of 0.96 ± 0.57, with all neurons except two having a value below 2 [17].

Mean values similar to those obtained by Volgushev *et al.* [17] were observed in the cortical data analyzed here, as shown in Table 1. However, consistent with the high value of the standard deviation in this analysis, 7 of the 27 RS cortical neurons had an average error ratio below 3 as seen in Fig 2, which would categorize them as having slow onsets. This low error ratio for these 7 RS cortical neurons can be attributed to the recording noise, which can bias the error ratio value lower toward one, and the selection of the range of the AP trace that is used for the fit [15]. Varying the data selection limits significantly impacts the error ratio of the neurons

analyzed in this study. Changing the upper limit from 30% of $\dot{V}_m$ maximum value to 3 to 10 mV above the onset potential increased the error ratio of the 7 previously miscategorized RS neurons, so that 6 out of the 7 have an error ratio higher than 3 using this alternative upper limit. The only neuron maintaining an error ratio below 3 after altering the upper limit is the neuron showing the slowest AP onset by all the quantification methods (RS neuron number 10 in Fig 2). A comparison of the impact of the data selection limits on two neurons is shown in S3 Fig. Furthermore, similar to the results from cortical neurons, the error ratios for the FS interneurons will shift from below 1 to above 3 as the portion of the AP data selected for fitting is varied. Changing the upper limit to be 3 mV above onset changed the error ratios for PVBCs to be 8.5 ± 2.8 (All values are presented as mean ± SD). While the other quantification methods do not depend on the data selection limits, the error ratio value can be significantly impacted by the limits of the data selection. Thus, the subsequent classification of a neuron as having slow or fast AP onset depends on the data selection when using the error ratio method.

## Classification of neuron types based on AP rapidity

**Differences in onset rapidity between RS and FS neurons.** Differences in the AP shapes of RS and FS cortical neurons, including quantitative features such as the AP width and amplitude, are well-known from the literature [5, 23, 26]. However, in prior studies, the AP onset rapidity has not been reported as one of the AP features used to differentiate RS from FS cortical neurons. Here, the results show that the AP onset rapidity is significantly different between RS and FS neurons, and hence provides another measure to differentiate between the two neuron classes. Based on mean rapidities of the above two populations of neurons as shown in Tables 1 and 3, FS cortical neurons have significantly higher rapidity than RS cortical neurons using all the quantification methods. However, the IFWd$^2$ and IHWd$^2$ methods have the highest scores of all methods in Table 3 according to all three statistics. The nonparametric, Mann-Whitney test, indicated all the AP quantification methods show a significant difference between RS and FS cortical neurons, as reflected in a z-score > 6 (Table 3).

The results from hippocampal neurons show similar patterns. For example, the FS PVBCs have AP onset that is 60% more rapid than pyramidal neurons based on analysis using the IFWd$^2$ method. However, the RS neurons mean rapidity was higher than the FS neurons mean rapidity in the hippocampus as quantified by the phase slope method using *spline* or *pchip* interpolation functions with a criterion level of 10 mV/ms and the error ratio method with the original upper limit set to 30% of the maximum $\dot{V}_m$. This relationship is denoted with a negative sign on all three statistical measures in Table 3, and reversed from what was found in the cortex, and thus expected in the hippocampus. Changing the upper limit on data selection for the error ratio method returned the mean FS rapidity to being larger than the mean RS rapidity in the hippocampus but resulted in degrading the effect size. Cohen's d, for comparison, decreased from large effect size of 2.47 to 0.36, a medium effect size. No reversal of FS and RS rapidity between cortex and hippocampus was observed for the IFWd$^2$ and IHWd$^2$ methods regardless of the interpolation technique.

In addition to AP rapidity, Table 3 summarizes how well AP amplitude, width, and onset potential differentiate RS and FS neuron recordings in this study. For both cortical and hippocampal neurons, a significant difference between RS and FS neurons is observed in all the AP parameters analyzed here except the AP onset potential between RS and FS hippocampal neurons. As shown in Table 1, the FS neurons, on average, have smaller AP amplitude, narrower AP width, lower onset voltage, and faster AP rapidity compared to the RS pyramidal neuron.

Together, results from the cortical and hippocampal neurons indicate that AP rapidity is clearly different between RS and FS neurons using parametric and nonparametric statistical

**Table 3. Comparison between electrophysiological properties of four neuron types using conventional statistics.**

| | Cortical | | | Hippocampal | | | Hippocampal and cortical pyramidal neurons | | |
| | FS and RS neurons | | | FS and RS neurons | | | | | |
| | t | z | Cohen's d (CLES) | t | z | Cohen's d (CLES) | t | z | Cohen's d (CLES) |
|---|---|---|---|---|---|---|---|---|---|
| **IFWd$^2$** ($ms^{-1}$) | 24.9 | 20.0 | 1.36 (0.84) | 49.6 | 25.3 | 2.77 (0.98) | 79.8 | 39.4 | 3.89 (1.0) |
| **IHWd$^2$** ($ms^{-1}$) | 16.2[e] | 15.6 | 0.97 (0.75) | 43.5 | 24.4 | 2.31 (0.97) | 55.6 | 35.6 | 2.89 (0.97) |
| **Phase Slope** ($ms^{-1}$) | 14.7 | 10.9 | 0.67 (0.70) | -16.0 | -24.9 | -0.64 (-0.71) | 17.7 | 39.1 | 0.98 (0.72) |
| | | | | -40.8[a] | -24.2[a] | -1.89[a] (-0.94[a]) | 58.1[c] | 39.5[c] | 3.18[c] (0.98[c]) |
| **Error ratio** (dimensionless) | 7.3 | 9.4 | 0.48 (0.62) | -59.4 | -25.5 | -2.47 (-0.98) | -3.8 | -6.2 | -0.14 (-0.54) |
| | | | | 5.3[b] | 6.4[b] | 0.36[b] (0.60[b]) | | | |
| **Amplitude** (mV) | -8.1 | -6.3 | -0.29 (-0.60) | -51.8[e] | -25.5 | -3.48 (-0.99) | 20.2 | 16.7 | 0.79 (0.72) |
| **Width** (ms) | -71.4 | -28.9 | -2.34 (0.98) | -133 | -25.8 | -5.54 (-1.0) | -40.2 | -36.1 | -1.46 (-0.87) |
| **Onset potential** (mV) | -23.3 | -18.6 | -1.16 (0.81) | 1.9[n] | 1.2[n] | 0.11 (0.53) | -17.6 | -14.8 | -0.64 (-0.69) |

Student's t test and Mann-Whitney test were performed to compare neuron types, and Cohen's d and common language effect size (CLES) were used to measure the effect size. For the t-score

[e] indicates that the equal variance hypothesis was accepted, however, the unequal variance t-score was within 4% within both cases.

[n] the difference is not significant; otherwise the difference is significant (p<0.05).

[a] using piecewise cubic interpolation.

[b] the upper limit was set to 3 mV above the onset.

[c] using piecewise cubic interpolation for hippocampal pyramidal neurons and cubic spline for cortical pyramidal neurons. The minus sign indicates that the RS neuron's mean was higher than the FS neuron's mean, and the cortical pyramidal neuron's mean was higher than hippocampal pyramidal neuron's mean.

tests (Table 3). The differences in rapidity using the $\ddot{V}_m$ peak width methods were not only significant as indicated by p-values but also have the largest effect size. As shown in Table 3, the only AP properties that produced an effect size larger than the IFWd$^2$ method in any of the comparisons is AP width in the intracortical and intrahippocampal comparisons and AP amplitude in the intrahippocampal comparison. AP width is known as providing a clear difference between RS and FS neurons [5, 26]. Both IFWd$^2$ rapidity and AP width score in the large effect size range. Therefore, the rapidity of AP onset can be used as another electrophysiological property to differentiate between these classes of neurons.

**Differences in onset rapidity between cortical and hippocampal pyramidal neurons.**
Pyramidal neurons are the most abundant excitatory neurons found in most mammalian forebrain areas such as the cerebral cortex and the hippocampus [27]. Pyramidal neurons in different areas have some family resemblance, but they vary in their morphology and behavior [28]. For example, a study showed that cortical pyramidal and CA1 hippocampal pyramidal neurons have similar Na$^+$ entry ratio and AP amplitude, but different AP width [29]. Therefore, differences between cortical and hippocampal pyramidal neuron AP onset rapidity, as well as the other AP parameters, are of interest.

AP onset rapidity is significantly different between cortical and hippocampal pyramidal neurons. Using the IFWd$^2$ method, the rapidity of hippocampal pyramidal neurons is more than double the rapidity of cortical pyramidal neurons. A very clear difference in rapidity is observed using all the quantification methods as shown in Table 3. Furthermore, comparing other electrophysiological properties revealed a significant difference between hippocampal and cortical pyramidal neurons in amplitude, onset potential, and width. The width of cortical neuron APs was more than 35% wider than those of the hippocampal neurons, which is typical [29–32]. Notably, the IFWd$^2$ method provided the highest significant difference among all the rapidity methods used in this analysis. Also, the $\ddot{V}_m$ peak width methods showed a significantly larger effect size compared to the other AP rapidity methods as shown in Table 3.

Double peaks in the rising phase of the $\ddot{V}_m$ trace, which have been associated with AP back-propagation in central mammalian neurons, were observed in a small minority of the analyzed APs and resulted in significantly lower rapidity than single-peaked $\ddot{V}_m$ traces in the rising phase. The double-component AP is characteristic of signal backpropagation from the AP initiation site [9, 17]. However, less than 7% of APs analyzed in this study had a clear shoulder or dip that corresponded to the double-component AP as examples given by Volgushev *et al* [17]. No FS neurons in the dataset exhibited a double-component AP. The rapidity of double-component APs was statistically significantly smaller than for single-component APs. The difference in rapidity of single and double component APs was most significant using the IFWd$^2$ method (cortex: Mann-Whitney Z = 12.8, p<0.0001, CLES = 0.79, hippocampus: Mann-Whitney Z = 23.2, p<0.0001, and CLES = 0.97). Similarly, the mean double-component rapidity found using the IHWd$^2$ and phase slope methods was less than the mean rapidity for single-component APs, but the level of significance and effect size was smaller or in some cases insignificant as shown in Table 4.

## Discussion

The results presented above not only support the $\ddot{V}_m$ peak width methods as improvements on existing methods of quantifying AP onset rapidity, but also support their utility in categorizing types of neurons. The $\ddot{V}_m$ peak width methods can capture the difference in AP rapidity between different neuron types while showing smaller relative variation across the same neuron type than prior methods. Therefore, the $\ddot{V}_m$ peak width methods can be used to classify different neuron types and hence enable quantitative analysis of factors impacting AP onset dynamics.

**Table 4. Comparison between single-component and double-component APs.**

| Neuron type | Cortical pyramidal neuron | | | | Hippocampal pyramidal neuron | | | |
|---|---|---|---|---|---|---|---|---|
| $\ddot{V}_m$ peaks | single | double | Z | Cohen's d (CLES) | single | double | Z | Cohen's d (CLES) |
| IFWd$^2$ (ms$^{-1}$) | 2.2 ± 0.5 | 1.5 ± 0.6 | 12.8* | 1.35 (0.79) | 4.6 ± 0.8 | 2.9 ± 0.4 | 23.2* | 2.17 (0.97) |
| IHWd$^2$ (ms$^{-1}$) | 4.0 ± 0.7 | 3.6 ± 1.7 | 0.95 | 0.58 (0.52) | 7.8 ± 1.8 | 3.9 ± 1.3 | 22.4* | 2.25 (0.96) |
| Phase Slope (ms$^{-1}$) | 7.0 ± 2.2 | 5.3 ± 2.9 | 6.5* | 0.73 (0.65) | 34.4 ± 16.9[a] | 32 ±13.5[a] | 0.93 | 0.14 (0.52) |

Z-score from Mann-Whitney test were used to compare AP waveform, Cohen's d and common language effect size (CLES) were used to measure the effect size.

[a] using piecewise cubic interpolation.

* the difference is significant (p<0.0001).

## The AP rapidity quantification methods

The results from the $IFWd^2$ and $IHWd^2$ methods are more reliable measures of AP rapidity than the other methods for two primary reasons. First, the points at which the rapidity values are calculated are well defined and don't require arbitrary choices of parameters. The $IFWd^2$ and $IHWd^2$ are measured at specific points on the $\ddot{V}_m$ trace, where the location of those points is defined using only their values relative to the peak value of $\ddot{V}_m$ without requiring an arbitrary or unscaled value or being influenced by extending the range of data analyzed. Unlike the $IFWd^2$ and $IHWd^2$ methods, the phase slope is measured at an arbitrary value on the phase plot, which differs from one study to another, while the error-ratio value depends on the portion of the AP recording selected for fitting. Thus, the $IFWd^2$ and $IHWd^2$ methods provide more consistent rapidity values that simplify comparison between different studies. Second, the second derivative peak width methods provide rapidity values that are independent of the definition of AP onset. The determination of AP rapidity, AP onset voltage variability, and the relationship between them are subjects of interest and great debate in many studies [6, 11, 15, 33]. The widely used phase slope method defines the AP onset voltage and rapidity at the same value of the phase slope. Shifting the criterion level at which the slope was measured was shown to have little effect in cortical neurons but can cause a large shift in the rapidity in computational models [10, 15]. However, shifting the criterion level can alter the results when the AP has a smooth onset [15], or when the $\dot{V}_m$ value is in close proximity to the kink. As a result, the phase slope value tripled when the criterion level was shifted from 10 mV/ms to 40 mV/ms. Furthermore, the comparison in Table 2 shows that the choice of interpolation function significantly influences the interpolated phase slope, leading to different rapidity values for many APs. While the onset voltage might not be significantly impacted by the choice of the interpolation function, the slope often is. The slope is more sensitive to excursions that can occur in the spline fit, while the onset voltage variability in this steep part of the phase space plot would be limited. As a result, the choice of the interpolation function could cause high variation in the phase slope that does not reflect the real differences in rapidity, which might in turn influence the analysis of the relationship between AP rapidity and threshold variability.

While the $IFWd^2$ and $IHWd^2$ are more reliable methods on good recordings, they are susceptible to noise, which is more pronounced when computing the second derivative. Noise can also introduce uncertainty in the peak position of the rising phase. Thus, analysis of a noisy recording requires the data to be filtered before calculating the $IFWd^2$ and $IHWd^2$. Nonetheless, the noise in the recordings analyzed here was quite small, and thus the calculation of the $IFWd^2$ and $IHWd^2$ was done without using any noise filters.

## Factors affecting AP onset rapidity

APs in central mammalian neurons have sharper and more abrupt onset compared to invertebrate neuron APs. An initial proposal to explain the "kink" in cortical neuron APs was introduced by Naundorf *et al* [6], where cooperative VGSCs gating was proposed to explain the sharp AP onset in cortical neurons. Naundorf *et al.* showed that the rapid AP onset and the variability of AP onset observed in cortical neurons can be replicated using a cooperative VGSC model instead of a canonical HH model, which failed to reproduced these two features [6]. However, an alternative explanation was introduced by Yu *et al.* [9] showing that the two features can be replicated using a multicompartment HH model without the need for cooperative VGSC gating, an explanation which was supported by patch-clamp recordings obtained from the soma of cortical neurons as well as from axonal bleb. They showed that the sharp somatic AP onset in cortical neurons is influenced by the distance from the axon initial segment (AIS) to the soma, with the rapidity increasing as the AP propagated away from the AIS

and become biphasic [9]. Therefore, AP rapidity can indicate distance between the somatic recording site and the AP initiation site, where neurons exhibiting lower AP onset rapidity indicate that the AP was initiated closer to the soma [34]. However, double-component APs were present in only a small portion of the APs analyzed here, and in those double-component APs, the rapidity was smaller compared to single-component APs, indicating that the AP back-propagation did not lead to an increase in neuron rapidity in this study. The analysis here agrees with Volgushev *et al's* [17] results where APs with double-components had a lower rapidity than single-component APs, although their results did not show a significant differ-ence between the two groups in cortical neurons. While AP backpropagation can contribute to the sharpness of the somatic AP, it was found that the AP backpropagation is necessary but not sufficient alone to reproduce the observed kink in cortical neurons [15].

Subsequent to publication of the AP backpropagation and cooperative gating theory, neu-ron geometry was proposed to explain the sharp AP rapidity observed in central mammalian neurons. In 2013, Romain Brette demonstrated theoretically that the abrupt AP onset observed in cortical neurons could be due to a different mechanism elucidated in resistive coupling the-ory [11]. The theory states that $Na^+$ current originating in the AIS is primarily sunk by the soma, due to its large size, and subsequently exits as capacitive current. Hence, the neuron geometry significantly influences the sharpness of the AP [10, 11, 35]. Other studies similarly confirmed the role of neuron size by demonstrating the large impact of dendritic tree size on AP rapidity [12, 36]. Eyal *et al.* [12] showed, using computational models, that rapidity increased by 30% when the axon-to-somatodendritic conductance ratio was increased from 12 to 370, and increased by 450% when the ratio was altered in the presence of ultrafast VGSC kinetics, which might also indicate the importance of specific VGSC subtypes. For example, while both the somatosensory cortex and hippocampus express similar VGSC subtypes, Nav1.1, Nav1.2, Nav1.3, and Nav1.6, the level of expression differs [37], which could be one of the factors contributing to the difference in rapidity between pyramidal neurons in the two regions. Moreover, another direct explanation of the difference in rapidity can be attributed to input resistance. For instance, the slower rapidity in cortical pyramidal neurons, compared to that in hippocampal pyramidal neurons, could be attributed to the higher input resistance of cortical pyramidal neurons [38, 39]. Nonetheless, these studies indicate the complexity of fac-tors mediating the rapidity and highlight the importance of a better and sensitive tool to mea-sure the impact of these factors on AP rapidity.

## The second derivative peak width methods for neuron classification

The ability to distinguish different neuron types is essential for understanding neuronal cir-cuits and functions. Cortical and hippocampal neurons have been classified based on different properties such as morphology, location, and electrophysiological properties. Here, as shown in Fig 5, the rapidity of AP onset can also be used to differentiate various neuron types. Analy-sis of the rapidity shows a significant difference between RS and FS neurons both in the somatosensory cortex and the CA1 hippocampus, and between cortical and hippocampal pyramidal neurons. These results agree with a previous study using the phase slope method that showed a significant difference in AP rapidity between two cell types, CA3 pyramidal neu-rons and dentate granule neurons [34]. Here, the $\ddot{V}_m$ peak width methods provide better sepa-ration between different cell types compared to the phase slope and the error ratio methods based on parametric and non-parametric statistical tests. The cortical and hippocampal recordings used in this study were obtained from two different research groups, and thus the recording and preparation conditions might contribute to the different rapidity between the two brain regions. However, the results from these datasets reproduced relationships of other

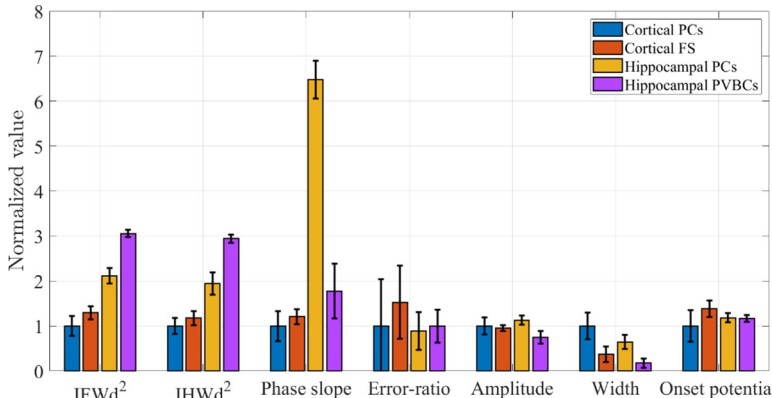

**Fig 5. Comparison of the electrophysiological properties of all the neurons analyzed in this study.** The vertical axis represents the values normalized to the mean value for cortical pyramidal neurons, and the error bar represents the RSD for each neuron population. The phase slope value for hippocampal pyrmidal cells is obtained using pchip interpolation, and the error ratio value for PVBCs are obtained with the upper limit was set to 3 mV above the onset.

AP parameters such as amplitude and width between cortical and hippocampal pyramidal neurons found in a previous study [29]. Carter and Bean [29] reported that cortical pyramidal neurons had wider and slightly higher amplitude APs than those recorded from hippocampal pyramidal neurons. While varying experimental conditions could influence conclusions about the relative rapidity of hippocampal and cortical pyramidal neurons, it would be expected to have a similar effect on all rapidity quantification methods. Therefore, the IFWd$^2$ method is expected to still better quantify any differences between hippocampal and cortical neuron rapidity than the other methods, as shown in Table 5.

## Sodium channel parameters affect modeled rapidity

Parameters of AP features have been correlated to differences in voltage-gated ion channel subtypes, such as the connection between VGKCs and AP width. In this study, the other AP feature that provides a good separation between the different neuron types is the AP width (S2 Table). The significant difference in the AP width between all the neuron types analyzed here is consistent with a difference in the types and densities of potassium channels expressed in these neurons since the AP width is mainly determined by VGKCs [2, 29, 40]. The significant

**Table 5. Two-tailed p-value from the t-score between the IFWd$^2$ values from different neuron types, and Cohen's d effect size (in parentheses).**

| IFWd$^2$ | | Cortex | | Hippocampus | |
|---|---|---|---|---|---|
| | | **PCs** | **FS** | **PCs** | **PVBCs** |
| **Cortex** | **PCs** | | <0.0001 | <0.0001 | <0.0001 |
| | | | (1.36) | (3.89) | (9.07) |
| | **FS** | <0.0001 | | <0.0001 | <0.0001 |
| | | (1.36) | | (2.52) | (8.13) |
| **Hippocampus** | **PCs** | <0.0001 | <0.0001 | | <0.0001 |
| | | (3.89) | (2.52) | | (2.77) |
| | **PVBCs** | <0.0001 | <0.0001 | <0.0001 | |
| | | (9.07) | (8.13) | (2.77) | |

Green cells indicate a p-value below 0.05.

"PCs" indicates pyramidal cells, and "PVBCs" indicate parvalbumin-positive basket cells.

difference in AP width is evidence of the key role of different VGKCs types whose activation and inactivation control many aspects of AP waveforms [2, 40–42].

In contrast to the variation in VGKCs associated with AP width, the activation of VGSCs has been proposed to be similar for hippocampal interneurons and their principle counterparts, pyramidal neurons [41]. Furthermore, some investigations have even shown that the gating properties of mammalian central neurons are similar to those in the giant squid [2]. Although several studies have shed light on the differences in VGSC kinetics between different neuron types, the role of VGSC activation in AP initiation is thought to be similar since the slope factor and time course of VGSCs was comparable [43]. Such conclusions agree with other studies in cortical neurons that show the rate of maximum rise was indistinguishable between RS and FS neurons [5, 26], indicating similar densities and behavior of sodium channel activation in both RS and FS cortical neurons. However, several other studies have argued that the fast AP onset observed in cortical neurons are due to cooperative VGSC activation, as evident by the high rapidity measured using the phase slope [6–8]. Whether the studies supported the role of VGSC activation, such as those proposing cooperative activity, or found no differences in VGSC activation, all these parameters were used to study VGSCs kinetics since the upstroke of APs is dominated by sodium current. Hence, it is reasonable to expect that the peak of the rising phase of the $\ddot{V}_m$ trace is dominated by sodium current. Moreover, in a recent study, we showed with a computational model that the $\ddot{V}_m$ peak width methods are more sensitive and specific to VGSC conductance and rate constant parameters than the phase slope method [18]. For instance, tripling the VGSC conductance caused a 65% increase in rapidity as quantified by the IFWd$^2$ method, but it caused only a l0% increase as quantified by the phase slope method [18]. Thus, the $\ddot{V}_m$ peak width methods are better tools to study the potential role of VGSCs in AP onset dynamics. These improved tools showed significant differences in AP rapidity between all 4 neuron types, albeit using recordings that may have had some different experimental procedures. Future analysis of RS and FS neuron recordings acquired from the cortex and hippocampus under comparable conditions using the $\ddot{V}_m$ peak width methods could further elucidate VGSC differences in these neurons.

## Conclusions

Two novel methods, IFWd$^2$ and IHWd$^2$, are more reliable and systematic tools to quantify the rapidity of AP onset than other existing methods. These $\ddot{V}_m$ peak width methods provide a smaller relative variation among APs from a single neuron type while still distinguishing between different neuron types more robustly than the phase slope and error ratio methods. Using the new second derivative methods, AP onset rapidity has been demonstrated as useful for neuron classification. Thus, the IFWd$^2$ and IHWd$^2$ tools should prove valuable for studying and analyzing the kinetics of VGSCs and their role in AP dynamics, such as examining different hypotheses proposed to cause the rapid AP initiation in central mammalian neurons.

## Supporting information

**S1 Fig. Pooled mean values for hippocampal pyramidal neuron rapidity.** The first 9 neurons are deep pyramidal neurons, while the following 8 neurons are superficial pyramidal neurons. Blue circle: Values using spline interpolation. Red diamond: Values using pchip interpolation.
(DOCX)

**S2 Fig. Pooled standard deviation values for hippocampal pyramidal neuron rapidity.** The first 9 neurons are deep pyramidal neurons, while the following 8 neurons are superficial

pyramidal neurons. Blue circle: Values using spline interpolation. Red diamond: Values using pchip interpolation.
(DOCX)

**S3 Fig. The effect on the error ratio of changing the upper and lower limits of data selection is compared for two neurons.** Top left: The error-ratio value when the upper limit was set at varying voltages above onset potential and the lower limit was 9 ms before onset. Top right: The error-ratio value when the upper limit was set at varying percentages of the maximum $\dot{V}_m$ and the lower limit was 9 ms before onset. Bottom left: The error-ratio value when the upper limit was set at varying absolute voltages above onset potential and the lower limit was 5 ms before onset. Bottom right: The error-ratio value when the upper limit was set at varying percentages of the maximum $\dot{V}$ and the lower limit was 5 ms before onset. Blue diamonds represent the error ratio for the neuron labeled AL 133, and orange circles represent the error ratio for the neuron labeled AL 215 [19].
(DOCX)

**S4 Fig. The impact of interpolation function on the shape and thus height of the $\ddot{V}_m$ peak.** Black circles represent the $\ddot{V}_m$ points calculated from the raw recordings before applying any interpolation functions. The dotted red line shows the $\ddot{V}_m$ trace after applying the quadratic regression interpolation function (pchip), while the dashed blue line shows the $\ddot{V}_m$ trace after applying the cubic spline interpolation function (spline).
(DOCX)

**S5 Fig. The impact of the onset criterion level on the phase slope for hippocampal neurons.** Blue circles show the mean phase slope value at different criterion levels for the hippocampal RS pyramidal neurons. Red squares show the mean phase slope value at different criterion levels for the hippocampal FS PVBCs. All APs that have maximum $\dot{V}_m$ less than 45 mV/ms were excluded. Note that the rapidity for RS and FS hippocampal neurons cross using the phase slope method.
(DOCX)

**S6 Fig. The impact of the onset criterion level on the phase slope for cortical neurons.** Blue circles show the mean phase slope value at different criterion levels for the cortical RS pyramidal neurons. Red squares show the mean phase slope value at different criterion levels for the cortical FS neurons. All APs that have maximum $\dot{V}_m$ less than 45 mV/ms were excluded.
(DOCX)

**S1 Table. Electrophysiological properties using the pooled mean and standard deviation.**
(DOCX)

**S2 Table. Two-tailed p-value from the t-score between the IFWd$^2$ values from different neuron types, and Cohen's d effect size (in parentheses).** Green cells indicate a p-value below 0.05.
(DOCX)

## Acknowledgments

The authors thank David Darevsky and Sierra Lear for their helpful comments on the manuscript. Ahmed A. Aldohbeyb acknowledges support from King Saud University for his PhD studies. Jozsef Vigh acknowledges support from Colorado State University, College of Veterinary Medicine and Biomedical Sciences Research Council Award.

## Author Contributions

**Conceptualization:** Ahmed A. Aldohbeyb, Kevin L. Lear.

**Data curation:** Ahmed A. Aldohbeyb.

**Formal analysis:** Ahmed A. Aldohbeyb, Kevin L. Lear.

**Investigation:** Ahmed A. Aldohbeyb, Jozsef Vigh, Kevin L. Lear.

**Methodology:** Ahmed A. Aldohbeyb, Kevin L. Lear.

**Project administration:** Kevin L. Lear.

**Software:** Ahmed A. Aldohbeyb.

**Supervision:** Jozsef Vigh, Kevin L. Lear.

**Validation:** Ahmed A. Aldohbeyb, Jozsef Vigh.

**Visualization:** Ahmed A. Aldohbeyb, Kevin L. Lear.

**Writing – original draft:** Ahmed A. Aldohbeyb, Kevin L. Lear.

**Writing – review & editing:** Ahmed A. Aldohbeyb, Jozsef Vigh, Kevin L. Lear.

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
