## [Decision Letter · Decision Letter 0]

22 Oct 2020

PONE-D-20-24488

New Methods for Quantifying Rapidity of Action potential Onset Differentiate Neuron Types

PLOS ONE

Dear Dr. Aldohbeyb,

Thank you for submitting your manuscript to PLOS ONE. After careful consideration, we feel that it has merit but does not fully meet PLOS ONE’s publication criteria as it currently stands. Therefore, we invite you to submit a revised version of the manuscript that addresses the points raised during the review process:

Classification performance would be directly related to the effect size of the test, which is not discussed, and ideally be demonstrated directly by training a classifier and then evaluating performance on untrained test data. The authors should at a minimum add calculations of effect size in terms of Cohen’s d, common language effect size, or both.

Implications for underlying mechanisms: “Contrary to published reports on the maximum rise rate, the difference in AP rapidity between neuron types indicates a significant difference in the activation kinetics of voltage-gated sodium channels (VGSCs).” (l38) This statement is not well supported – elsewhere in the manuscript the authors mention the significant discussion in the literature over whether non-channel factors (passive interactions between compartments) contribute to differences in onset. They should therefore also acknowledge that these factors could provide alternative explanations for cross-type differences. (Input resistance differences are mentioned briefly as a possible factor on l473, but not elsewhere.)

In addition, when discussing how their results fit with the existing literature, it would be helpful for the authors to distinguish between the backpropagation [7] and critical resistive coupling [9, 14] theories, which are both alternative explanations for variations in AP onset not linked to sodium channel properties

Action potential backpropagation and, more precisely, the dendritic shunting, mentioned in the Yu, Shu and MacCormick paper of 2008 and investigated in detail by Eyal, Mansvelder, Kock and Segev (2014), are, probably, more important factors determining the rapidity of the action potential onset. Therefore, both the introduction and, especially, the discussion should be modified to reflect this fact. In addition, the last sentence in the Abstract should be modified accordingly.

The sample sizes that contain only 3 neurons of a particular type (CA1 hippocampal interneurons in Results section on page 17 – 18, Table 4 on page 21, the table is mislabeled as Table 1) are too small. It is not appropriate to use the t-test for such small groups because there is a substantial probability that all 3 measurements of the rapidity in these 3 neurons fall on one side of the distribution of the measure. Moreover, it is impossible to deduce what type of distribution is in case of such a small sample. Non-parametric statistical tests such as the Kruskal-Wallis test, which can be used also for a non-normal distribution, will show that such small groups are insufficient to make any conclusions. Therefore, either additional neurons should be included in these groups or all Results section describing differences between these groups and the corresponding Discussion section should be deleted.

For the comparison of hippocampal to cortical cells, the authors should address whether factors in addition to intrinsic differences could be contributing to the cross-type distinctions shown. Specifically, recording conditions likely differed between the two data sources, and should be discussed.

On lines 65 – 67 references are cited inappropriately. Reference 8 does not talk about ‘cooperative gating’: in this paper a simple 6 state model without any cooperation fits well sodium current traces that were found to be without a HH-type delay. In reference 7 backpropagation of action potentials alone is considered to be sufficient to account for the experimental data and the ‘cooperative’ gating model was found to be unnecessary. Please modify this sentence accordingly.

The numbering of tables is confused in several places:

- in p. 3 probably should be referred to the Table 1 but not Table 3;

- Table 1 is probably Table 4 (p. 21), this table is referred as Table 1 on page 20.

We look forward to receiving your revised manuscript.

Kind regards,

Gennady Cymbalyuk, Ph.D.

Academic Editor

PLOS ONE

Journal Requirements:

2. Please include a copy of Tables 4 and 5 which you refer to in your text.

Reviewers' comments:

Reviewer's Responses to Questions

**Comments to the Author**

1. Is the manuscript technically sound, and do the data support the conclusions?

Reviewer #1: Partly

Reviewer #2: Partly

2. Has the statistical analysis been performed appropriately and rigorously? 

Reviewer #1: No

Reviewer #2: I Don't Know

3. Have the authors made all data underlying the findings in their manuscript fully available?

Reviewer #1: Yes

Reviewer #2: Yes

4. Is the manuscript presented in an intelligible fashion and written in standard English?

Reviewer #1: Yes

Reviewer #2: Yes

5. Review Comments to the Author

Reviewer #1: The rate of information transfer in the brain depends on the rapidity of action potential onset, thus it is important to have a reliable measure of this parameter. The authors compare 4 different methods of measurement of this parameter and demonstrate that their proposed two methods, both based on the measurement of the peak width of the second derivative, are the most reliable.

Major concern: the authors went too far by analyzing very small samples that contain only 3 neurons of a particular type (CA1 hippocampal interneurons in Results section on page 17 – 18, Table 4 on page 21, the table is mislabeled as Table 1). It is not appropriate to use the t-test for such small groups because there is a substantial probability that all 3 measurements of the rapidity in these 3 neurons fall on one side of the distribution of the measure. Moreover, it is impossible to deduce what type of distribution is in case of such a small sample. Non-parametric statistical tests such as the Kruskal-Wallis test, which can be used also for a non-normal distribution, will show that such small groups are insufficient to make any conclusions. Therefore, either additional neurons should be included in these groups or all Results section describing differences between these groups and the corresponding Discussion section should be deleted.

Minor comments:

It is mentioned in several places that the rapidity of action potential onset reflects the kinetics of voltage gated sodium channels. This is true only in part. As it is mention in the introduction, action potential backpropagation and, more precisely, the dendritic shunting, mentioned in the Yu, Shu and MacCormick paper of 2008 and investigated in detail by Eyal, Mansvelder, Kock and Segev (2014), are, probably, more important factors determining the rapidity of the action potential onset. Therefore, both the introduction and, especially, the discussion should be modified to reflect this fact. In addition, the last sentence in the Abstract should be modified accordingly.

On lines 65 – 67 references are cited inappropriately. Reference 8 does not talk about ‘cooperative gating’: in this paper a simple 6 state model without any cooperation fits well sodium current traces that were found to be without a HH-type delay. In reference 7 backpropagation of action potentials alone is considered to be sufficient to account for the experimental data and the ‘cooperative’ gating model was found to be unnecessary. Please modify this sentence accordingly.

The numbering of tables is confused in several places:

- in p. 3 probably should be referred to the Table 1 but not Table 3;

- Table 1 is probably Table 4 (p. 21), this table is referred as Table 1 on page 20.

Reviewer #2: Major points

Differentiation vs classification: (l36-37, l381-385, etc) the authors emphasize the interesting result that the new methods better differentiate between neuron types, in the sense of significant pairwise t-tests between groups. However, this is necessary but not sufficient for the feature to be useful for classification, which is also claimed but not demonstrated (“for the first time indicating that AP rapidity can be used to classify neuron types”). Classification performance would be directly related to the effect size of the test, which is not discussed, and ideally be demonstrated directly by training a classifier and then evaluating performance on untrained test data. The authors should at a minimum add calculations of effect size in terms of Cohen’s d, common language effect size, or both.

Implications for underlying mechanisms: “Contrary to published reports on the maximum rise rate, the difference in AP rapidity between neuron types indicates a significant difference in the activation kinetics of voltage-gated sodium channels (VGSCs).” (l38) This statement is not well supported – elsewhere in the manuscript the authors mention the significant discussion in the literature over whether non-channel factors (passive interactions between compartments) contribute to differences in onset. They should therefore also acknowledge that these factors could provide alternative explanations for cross-type differences. (Input resistance differences are mentioned briefly as a possible factor on l473, but not elsewhere.)

In addition, when discussing how their results fit with the existing literature, it would be helpful for the authors to distinguish between the backpropagation [7] and critical resistive coupling [9, 14] theories, which are both alternative explanations for variations in AP onset not linked to sodium channel properties

Statistical analysis:

Use of pooled mean: In the case of statistics by type, the use of a pooled mean weighted by number of spikes seems a bit odd, since the primary source of variation is likely to be at the cell level, not the spike level. The pooled SD is therefore certainly not a good estimate of the variability of the population mean, and the population mean itself will be biased towards cells with more spikes. Finally, it is unclear if these pooled calculations are also used in the t-tests, which would certainly overstate the differences between types.

T-tests: the authors should justify whether the equal variance assumption of Student’s t-test is met here, or use a different test.

Minor clarifications:

When discussing the impacts of the interpolation, the following statement is not sufficiently clear: “significant reduction in the phase slope values is expected since the slope is calculated at a specific value (at 10 mV/ms) of the dependent (vertical axis) variable in phase space plots.”

For the comparison of hippocampal to cortical cells, the authors should address whether factors in addition to intrinsic differences could be contributing to the cross-type distinctions shown. Specifically, recording conditions likely differed between the two data sources, and should be discussed.

6. PLOS authors have the option to publish the peer review history of their article (what does this mean?). If published, this will include your full peer review and any attached files.

Reviewer #1: **Yes: **Gytis Baranauskas

Reviewer #2: **Yes: **Thomas Chartrand

---

## [Author Response · Author response to Decision Letter 0]

7 Dec 2020

Dear Dr. Cymbalyuk,

 We thank you and the reviewers for the comments on our manuscript, "New Methods for Quantifying Rapidity of Action potential Onset Differentiate Neuron Types". We revised the paper to address the editor and the reviewers' comments and edited the manuscript accordingly. We believed the suggested statistical test and discussion have made the paper stronger. On the following pages, you will find our response to the editor's and reviewers' comments.

 On behalf of the co-authors, I thank you for considering our paper for resubmission. 

Thanks

Ahmed Aldohbeyb

The sample sizes that contain only 3 neurons of a particular type (CA1 hippocampal interneurons in Results section on page 17 – 18, Table 4 on page 21, the table is mislabeled as Table 1) are too small. It is not appropriate to use the t-test for such small groups because there is a substantial probability that all 3 measurements of the rapidity in these 3 neurons fall on one side of the distribution of the measure. Moreover, it is impossible to deduce what type of distribution is in case of such a small sample. Non-parametric statistical tests such as the Kruskal-Wallis test, which can be used also for a non-normal distribution, will show that such small groups are insufficient to make any conclusions. Therefore, either additional neurons should be included in these groups or all Results section describing differences between these groups and the corresponding Discussion section should be deleted.

We excluded all neurons with a sample size equal to three. Such a decision was made after performing the Kruskal-Wallis test. The test showed that the combination of sample sizes between neurons with small and large sample size did not allow to find a critical value. Thus, we removed neuron types with a small sample size since the data were obtained from open-source databases, and we could not increase the sample size. However, we were able to perform the Kruskal-Wallis test between the three hippocampal FS interneuron types, which include 6 PVBCs, 3 BiCs, and 3 AACs. The test revealed that the difference between the three hippocampal FS interneuron types is significant. Nonetheless, we agree that the sample size of BiCs and AACs is small and might differ with larger sample size. Therefore, we excluded these neurons from the main manuscript, which now contain the results from 4 neuron types, and put the results for comparing FS interneuron types in the supporting information. 

Use of pooled mean: In the case of statistics by type, the use of a pooled mean weighted by number of spikes seems a bit odd, since the primary source of variation is likely to be at the cell level, not the spike level. The pooled SD is therefore certainly not a good estimate of the variability of the population mean, and the population mean itself will be biased towards cells with more spikes. Finally, it is unclear if these pooled calculations are also used in the t-tests, which would certainly overstate the differences between types. T-tests: the authors should justify whether the equal variance assumption of Student's t-test is met here, or use a different test.

We choose the pooled statistics because we believed cells within a specific cell type taken from the same region should have similar electrophysiological properties. However, the reviewer raised a valid argument. So, we add another way to calculate the mean and SD. Besides the pooled statistics, the mean and SD was calculated for all the APs obtained from a specific cell type. We believe that we provided more conservative tests to compare the different quantification methods with both the overall and pooled mean and SD. Also, we performed t-statistics and Cohen's d test for the results from the pooled and unpooled mean and SD. As expected, with the overall mean and SD values, the variation within a specific cell type was higher than the RSD obtained from the pooled statistics, but still, the RSD was smaller with the V ¨_m peak width methods compared to the other methods. For the differentiation ability, the IFWd2 method was the only rapidity quantification method to differentiate between RS and FS cortical neurons, as evident by the high t-score and the correspondent p-value, and the large effect size (Table 1 and Table S1). 

Classification performance would be directly related to the effect size of the test, which is not discussed, and ideally be demonstrated directly by training a classifier and then evaluating performance on untrained test data. The authors should at a minimum add calculations of effect size in terms of Cohen's d, common language effect size, or both.

Non-parametric and effect size tests were performed to assist the different AP rapidity quantification methods. Cohen's d effect size test was performed to both overall and pooled mean and SD. Then, the non-parametric Mann-Whitney U test was performed to compare different cell types. The Mann-Whitney U test was chosen because it was the test used by many of the classical classification papers such as McCormick et al. (1985) and Nowak et al. (2003). The results from the Mann-Whitney U test showed that all the rapidity quantification methods, except the error ratio in some cases, can differentiate between neuron types. Nonetheless, the IFWd2 method still had the highest effect size, t-score, and z-score (Table S1-S2).

Action potential backpropagation and, more precisely, the dendritic shunting, mentioned in the Yu, Shu and MacCormick paper of 2008 and investigated in detail by Eyal, Mansvelder, Kock and Segev (2014), are, probably, more important factors determining the rapidity of the action potential onset. Therefore, both the introduction and, especially, the discussion should be modified to reflect this fact. In addition, the last sentence in the Abstract should be modified accordingly.

For the comparison of hippocampal to cortical cells, the authors should address whether factors in addition to intrinsic differences could be contributing to the cross-type distinctions shown. Specifically, recording conditions likely differed between the two data sources, and should be discussed.

A detailed discussion was added to address the factors that could cause the difference between neuron types in AP rapidity. As shown in the second half of the discussion (line 526-580), we discussed the different mechanisms suggested in the literature to explain the difference in AP rapidity such as the AP backpropagation, somatodendritic size, and cooperative gating. Also, we acknowledge that some of the differences between cortical and hippocampal neurons could be due to differences in recording and preparation conditions (line 478-481). 

Implications for underlying mechanisms: "Contrary to published reports on the maximum rise rate, the difference in AP rapidity between neuron types indicates a significant difference in the activation kinetics of voltage-gated sodium channels (VGSCs)." (l38) This statement is not well supported – elsewhere in the manuscript the authors mention the significant discussion in the literature over whether non-channel factors (passive interactions between compartments) contribute to differences in onset. They should therefore also acknowledge that these factors could provide alternative explanations for cross-type differences. (Input resistance differences are mentioned briefly as a possible factor on l473, but not elsewhere). In addition, when discussing how their results fit with the existing literature, it would be helpful for the authors to distinguish between the backpropagation [7] and critical resistive coupling [9, 14] theories, which are both alternative explanations for variations in AP onset not linked to sodium channel properties

Done. We added a detailed discussion and acknowledge the factors that could contribute to the difference in AP rapidity.

For the statement ("Contrary to published reports…"), we think we did not communicate well what we meant. Thus, we changed the sentence, but we would like to explain our point of view. We acknowledge that in biology, with all the complexity, it is difficult to find a direct cause and effect relationship, which it is obvious as you stated from the different mechanisms that could explain the results. Especially, since we did not perform any experiments to support the statement such as applying TTX to see how the rapidity change with the available VGSCs. Yet, even in the classical neuron classification papers, the authors made some claim about the VGSCs kinetics based on different electrophysiological properties. For instance, the maximum rise rate was one of the electrophysiological properties that linked to VGSCs density. Nowak et al., (2003), for example, suggested that the significant difference in maximum rise rate between chattering-firing neurons and the other classes of cortical neurons is due to difference in VGSCs density. Thus, the comparable values , in Nowak et al., (2003) study, for RS and FS cortical neurons indicate similar VGSCs density. However, previous studies have shown that the relationship between maximum rise rate and maximum sodium conductance is not linear and exclusive. For example, Sheets, Hanck, and Fozzard (1987) stated that the rate of rise is not a reliable method to indicate maximum sodium conductance. By the same token, However, we believe that the IFWd2 and IHWd2 methods are better tool to study VGSCs since the rising phase of the V ¨_m trace is dominated by sodium current. We supported this argument, in a previous study, using an HH model by showing that the V ¨_m peak width methods have high sensitivity and specificity to VGSCs kinetics and density compared to the phase slope method (ref. 18). For example, we showed that when we increased the VGSCs five-fold, while keeping VGKCs the same, the IFWd2 and IHWd2 values doubled. On the other hand, the phase slope value increased by less than 14%. Therefore, although we acknowledge that changes in AP rapidity might not be linked solely to VGSCs kinetics or maybe not linked at all, we believe that the V ¨_m peak width methods are better a tool to study the impact of VGSCs on AP dynamics than the ones used in previous studies. 

On lines 65 – 67 references are cited inappropriately. Reference 8 does not talk about 'cooperative gating': in this paper a simple 6 state model without any cooperation fits well sodium current traces that were found to be without a HH-type delay. In reference 7 backpropagation of action potentials alone is considered to be sufficient to account for the experimental data and the 'cooperative' gating model was found to be unnecessary. Please modify this sentence accordingly.

Done. The references were corrected (line 66-71). 

The numbering of tables is confused in several places:

- in p. 3 probably should be referred to the Table 1 but not Table 3;

- Table 1 is probably Table 4 (p. 21), this table is referred as Table 1 on page 20.

Done. Table 1 was moved to page 9, and table 2 was deleted since hippocampal neuron types were deleted from the manuscript and table 1 summarizes the results from all the remaining 4 neuron types. 

When discussing the impacts of the interpolation, the following statement is not sufficiently clear: “significant reduction in the phase slope values is expected since the slope is calculated at a specific value (at 10 mV/ms) of the dependent (vertical axis) variable in phase space plots.”

We clarify the sentence (see line 259-261).

---

## [Decision Letter · Decision Letter 1]

4 Jan 2021

PONE-D-20-24488R1

New Methods for Quantifying Rapidity of Action Potential Onset Differentiate Neuron Types

PLOS ONE

Dear Dr. Aldohbeyb,

Thank you for submitting your manuscript to PLOS ONE. After careful consideration, we feel that it has merit but does not fully meet PLOS ONE’s publication criteria as it currently stands. Therefore, we invite you to submit a revised version of the manuscript that addresses the points raised during the review process.

The manuscript should be revised for readability.

Abbreviations in tables should be used more consistently: Somatosensory cortex in Table 1, S.S. Cortex in Table 3 and SS cortex in Table 4.

On line 332 ‘methods have the highest values on all three statistics’ probably should read ‘methods have the highest power according to all statistical tests used’.

It could be useful to have a list of abbreviations, some of them are difficult to interpret and may be confusing, for instance VGKC and VGSC.

The new analysis comparing single and double-component could be shortened and made more readable by focusing on a single metric, probably IFWD, and presenting the details of which comparisons are significant in a table.

The common language effect size for the Mann-Whitney test (equivalent to the area-under-curve from ROC analysis) might be a more relevant effect size for situations in which the non-parametric MW test is considered the primary test indicated by the properties of the data.

We look forward to receiving your revised manuscript.

Kind regards,

Gennady Cymbalyuk, Ph.D.

Academic Editor

PLOS ONE

Reviewers' comments:

Reviewer's Responses to Questions

**Comments to the Author**

1. If the authors have adequately addressed your comments raised in a previous round of review and you feel that this manuscript is now acceptable for publication, you may indicate that here to bypass the “Comments to the Author” section, enter your conflict of interest statement in the “Confidential to Editor” section, and submit your "Accept" recommendation.

Reviewer #1: (No Response)

Reviewer #2: (No Response)

2. Is the manuscript technically sound, and do the data support the conclusions?

Reviewer #1: Yes

Reviewer #2: Yes

3. Has the statistical analysis been performed appropriately and rigorously? 

Reviewer #1: Yes

Reviewer #2: Yes

4. Have the authors made all data underlying the findings in their manuscript fully available?

Reviewer #1: Yes

Reviewer #2: Yes

5. Is the manuscript presented in an intelligible fashion and written in standard English?

Reviewer #1: Yes

Reviewer #2: Yes

6. Review Comments to the Author

Reviewer #1: Although the authors adequately addressed all issues raised in my comments there are still language problems in the text that should be addressed. First, abbreviations in tables should be used more consistently: Somatosensory cortex in Table 1, S.S. Cortex in Table 3 and SS cortex in Table 4. Second, on line 332 ‘methods have the highest values on all three statistics’ probably should read ‘methods have the highest power according to all statistical tests used’. Third, it could be useful to have a list of abbreviations, some of them are difficult to interpret and may be confusing, for instance VGKC and VGSC. Finally, the manuscript should be made more readable by somebody with more experience in paper writing of by professional editing services, there are many places that are difficult to follow.

Reviewer #2: The authors’ revisions are appreciated, and have substantially increased the quality of the manuscript. I support publication in the current form, but have a few minor, non-essential edits to suggest.

The addition of the “Factors affecting AP onset rapidity” section in the discussion is very useful. It could be additionally improved by minor edits noting which developments are supported by computational modeling vs experimental evidence.

Likewise, the new analysis comparing single and double-component APs adds significantly to the links to this prior work in the field. It could, however, be shortened and made more readable by focusing on a single metric, probably IFWD, and presenting the details of which comparisons are significant in a table.

Finally, although I think the broad presentation of alternative statistics in the current version is more than sufficient, I’d like to note that the common language effect size for the Mann-Whitney test (equivalent to the area-under-curve from ROC analysis) might be a more relevant effect size for situations in which the non-parametric MW test is considered the primary test indicated by the properties of the data.

7. PLOS authors have the option to publish the peer review history of their article (what does this mean?). If published, this will include your full peer review and any attached files.

Reviewer #1: **Yes: **Gytis Baranauskas

Reviewer #2: **Yes: **Thomas Chartrand

---

## [Author Response · Author response to Decision Letter 1]

12 Jan 2021

reviewer #1: Although the authors adequately addressed all issues raised in my comments there are still language problems in the text that should be addressed. First, abbreviations in tables should be used more consistently: Somatosensory cortex in Table 1, S.S. Cortex in Table 3 and SS cortex in Table 4. Second, on line 332 'methods have the highest values on all three statistics' probably should read 'methods have the highest power according to all statistical tests used'. Third, it could be useful to have a list of abbreviations, some of them are difficult to interpret and may be confusing, for instance VGKC and VGSC. Finally, the manuscript should be made more readable by somebody with more experience in paper writing of by professional editing services, there are many places that are difficult to follow.

We thank the reviewer for their insightful comments. We apologize for the inconsistent abbreviations in the table. We fixed these issues and added a list of abbreviations on the first page. Also, as suggested, the sentence in line 332 was modified, and the manuscript was revised for readability and was then reviewed and revised by the editor-in-chief for the Berkeley Science Review. 

Reviewer #2: The authors' revisions are appreciated, and have substantially increased the quality of the manuscript. I support publication in the current form, but have a few minor, non-essential edits to suggest.

The addition of the "Factors affecting AP onset rapidity" section in the discussion is very useful. It could be additionally improved by minor edits noting which developments are supported by computational modeling vs experimental evidence. Likewise, the new analysis comparing single and double-component APs adds significantly to the links to this prior work in the field. It could, however, be shortened and made more readable by focusing on a single metric, probably IFWD, and presenting the details of which comparisons are significant in a table. Finally, although I think the broad presentation of alternative statistics in the current version is more than sufficient, I'd like to note that the common language effect size for the Mann-Whitney test (equivalent to the area-under-curve from ROC analysis) might be a more relevant effect size for situations in which the non-parametric MW test is considered the primary test indicated by the properties of the data.

We thank the reviewer for their insightful comments. The discussion section on factors affecting AP onset rapidity was updated to indicate the type of evidence used in the cited studies. For the analysis comparing AP waveforms with single and double components, we shortened the section and summarized the results in a table instead, as suggested. Finally, we used the common language effect size test in all comparisons as shown in Tables 3 and 4.

---

## [Decision Letter · Decision Letter 2]

4 Feb 2021

New Methods for Quantifying Rapidity of Action Potential Onset Differentiate Neuron Types

PONE-D-20-24488R2

Dear Dr. Aldohbeyb,

We’re pleased to inform you that your manuscript has been judged scientifically suitable for publication and will be formally accepted for publication once it meets all outstanding technical requirements.

Kind regards,

Gennady Cymbalyuk, Ph.D.

Academic Editor

PLOS ONE

Additional Editor Comments (optional):

Reviewers' comments:

Reviewer's Responses to Questions

**Comments to the Author**

1. If the authors have adequately addressed your comments raised in a previous round of review and you feel that this manuscript is now acceptable for publication, you may indicate that here to bypass the “Comments to the Author” section, enter your conflict of interest statement in the “Confidential to Editor” section, and submit your "Accept" recommendation.

Reviewer #1: All comments have been addressed

2. Is the manuscript technically sound, and do the data support the conclusions?

Reviewer #1: Yes

3. Has the statistical analysis been performed appropriately and rigorously? 

Reviewer #1: Yes

4. Have the authors made all data underlying the findings in their manuscript fully available?

Reviewer #1: Yes

5. Is the manuscript presented in an intelligible fashion and written in standard English?

Reviewer #1: Yes

6. Review Comments to the Author

Reviewer #1: The authors addressed all issues raised and I have no further comments. I believe that the manuscript is suitable for publications in its current form.

7. PLOS authors have the option to publish the peer review history of their article (what does this mean?). If published, this will include your full peer review and any attached files.

Reviewer #1: **Yes: **Gytis Baranauskas

---

## [Editor Report · Acceptance letter]

1 Apr 2021

PONE-D-20-24488R2 

New Methods for Quantifying Rapidity of Action Potential Onset Differentiate Neuron Types 

Dear Dr. Aldohbeyb:

I'm pleased to inform you that your manuscript has been deemed suitable for publication in PLOS ONE. Congratulations! Your manuscript is now with our production department. 

Kind regards, 

on behalf of

Dr. Gennady S. Cymbalyuk 

Academic Editor

PLOS ONE